# Designing Heat-Set Gels for Crystallizing APIs at Different Temperatures: A Crystal Engineering Approach

Pathik Sahoo [1,2]

1    Foundation of Physics Research Center (FoPRC), 87053 Cosenza, Italy; 2c.pathik@gmail.com
2    International Institute of Invincible Rhythms, Churat Nala Road, Chakrail, Shimla 171006, India

**Abstract:** An organic salt crystallizes through different kinds of charge-assisted hydrogen-bonded networks depending on carboxylic functionality number and the degree of amine. These H-bonded packing patterns are often robust and predictable, so one can design a supramolecular salt with a certain purpose. In some cases, two different crystalline packing patterns can be found in Primary Ammonium Dicarboxylate (PAD) salts at different temperatures. Two kinds of supramolecular bonding, namely, charge-assisted hydrogen bonding and weak van der Waals interactions stabilize the two states. A small increase in the carbon chain length in a primary amine enhances the additional van der Waals interactions with the packing so that the 2D hydrogen-bonded network (HBN) transforms into a 1D HBN at room temperature. Such van der Waals interactions can be controlled by external heat, so a temperature-dependent 1D to 2D phase change is feasible. When certain moieties, such as azo and bipyridine, are introduced into the carboxylic acid backbone, the acids become insoluble in most organic solvents, raising their melting point, and resulting in heat-set gels. In the presence of an API, temperature and solvent-dependent polymorphic crystals can be grown in the heat-set gel medium and by simply cooling down the mixture, the API crystals can be separated easily.

**Keywords:** supramolecular synthon; bio-active polymorphs; phase transition; specific heat; supersaturation





## 1. Introduction

A general strategy for designing a robust supramolecular structure is often based on the topology of the constituent molecules/ions [1–3], the specific functional group on the molecular backbone [4], and the length between the functional groups [5]. However, physical properties such as the melting point of a single constituent molecule in a multi-component supramolecular system have never been investigated in tuning supramolecular properties.

Supramolecular gelators are small organic molecules of molecular weight below 3000 that can form self-assembly in an appropriate solvent above the minimum gelator concentration (MGC) to form supramolecular gel [4]. The gelators can form 1D, 2D, or 3D networks depending on molecular functionalities. The 1D self-assembled networks are frequently observed in gel, where they appear fiber-like and entangle together through junction jones to form a 3D cage to trap the gelling solvent by viscoelastic forces. By forming a 2D self-assembly, a supramolecular network forms a plate-type morphology [6]. These plates form a clumsy 3D aggregate through irregular and poor stacking to seize the gelling solvent. Finally, 3D networks are generally found in 3D coordination polymer gels [3] where the solvent molecules are seized inside the supramolecular cavity. Irrespective of the packing pattern, the supramolecular gels are thermo-reversible in nature. When a supramolecular gelator and its corresponding gelling solvent are heated together, it forms a solution (sol) and while cooling it forms a gel [7,8]. The interconversion of the sol and gel state by conducting the heating and cooling cycles establishes the thermo-reversible property in a gel. The formation of gels during the cooling cycle is abundant and categorized as a thermally reversible supramolecular gel. In contrast, the gel formed during the heating cycles is designated as heat-set gel or reverse thermal gel. Just the tuning of

the melting point of dicarboxylic acid in PAD salt can convert a normal thermo-reversible gel [7,8] into a heat-set gel [6].

A series of supramolecular gelators, derived through Primary Ammonium Dicarboxylate (PAD) synthon [6–8], can generate either a heat-set gel (reverse thermal gel) [6,9] or thermally reversible supramolecular gel [8] just by tuning the melting point of a constituent, di-carboxylic acid. The PAD salts can be made by reacting a dicarboxylic acid with long-chain primary amine molecules in 1:2 molar ratios in a suitable solvent. In the presence of an appropriate solvent, PAD salts can form either a 1D or 2D network to form supramolecular gels. This PAD salt crystallizes in two polymorphs, one with a one-dimensional hydrogen-bonded network (HBN) and the other with a two-dimensional HBN [8,10]. By matching the simulated pattern observed from the single crystal structure of gelators with the gel or xerogels, we can indirectly predict the supramolecular packing pattern in the gel [7,8]. Furthermore, the 1D network-derived gel fiber and 2D network-derived thin plate in supramolecular packing patterns can be observed under SEM. In the solid state, these compounds both have charge-aided hydrogen bonds (−14 to −17 Kcal/mol) [11] and van der Waals interactions (~ 1 Kcal/mol) [12]. Hydrogen bonding is directional and develops 3D spatial interactions between carboxylate and ammonium moieties in forming a 1D/2D supramolecular network. However, despite a noticeable weakness in van der Waals interactions, it controls the packing pattern collectively in the solid-state [12]. With increasing amine chain length, the enhanced van der Waals interaction transforms the 2D network into the 1D network [10]. By raising the temperature, the PAD slats can undergo a crystalline phase change and produce heat-set gels, if the dicarboxylic acid does not melt and become solubilized in gelling solvents (Figure 1a). It should be noted that, by replacing the primary amine with secondary amines, Secondary Ammonium Dicarboxylate (SAD) synthone can be produced [13]. These SAD synthons are extremely robust in nature and no polymorphs can be grown as a result; this can not be used in designing heat-set gelator or mesogenic gelator molecules (Figure 1b).

As a higher collective van der Waals energy [12] through the alkyl chain makes a 2D HBN into 1D HBN [8,10], the temperature might be an interesting candidate in tuning the supramolecular structure rather than the same salt. The advantage of temperature is that it can tune a single supramolecular structure, but the van der Waals interaction cannot produce two packing patterns in a given system at a given temperature. A simple differential scanning calorimetry (DSC) [6] or different temperature-based X-ray powder diffraction (XRPD) experiment can depict the different crystal phases in the supramolecular structure.

When ferrocene dicarboxylic acid was used to obtain an organometallic helical gel [14] by PAD synthon [7], it was noted that the series of salts produced a weak gel-like material after heating the gelator in aromatic solvents but produced a colloid at room temperature. These were wrongly reported as weak gels, rather than reported as heat-set gels. In the next project, the PAD salts derived from the azobenzene-4,4′-dicarboxylic acid and primary amines exhibited the reverse thermal gelling property [6] where the rheological data showed that the gelling strength increased with the increase in the carbon chain length of the primary amine. Two dicarboxylic acids, namely ferrocene dicarboxylic acid and azobenzene-4,4′-dicarboxylic acid, do not melt in the heating, rather they decompose at around 350 °C. If a structural part of the gelator makes the gelator insoluble in gelling solvent and does not melt even in a solution, the thermo-reversible gelator is converted into a reverse thermal gelator or heat-set gelator.

In addition to organic salts, coordination polymers, biological polymers, and charge transfer complexes can also exhibit the heat-set gelation property. Polymeric compounds impart gelation of their large molecular weight, which cannot be dissolved in a high-temperature medium. For example, the coordination polymer based on 1,2,4-triazole ligand coordinated Co(II) derived polymer gel shows the heat-set gel from tetrahedral coordination geometry to octahedral coordination geometry. The lipophilic in that 1D polymer provides unique properties in designing a thermo-responsive gel [15]. A heat-set gelator can also be designed by introducing a long chain at the coordination polymer [16]. Some biological

polymer materials such as Globular Protein Gel [17], Polyisocyanopeptide Polymers [18], quinoa protein [19], RuBisCO [20], konjac glucomannan [21], and polyurethane/PEG block copolymer [22] also form a heat-set gel.

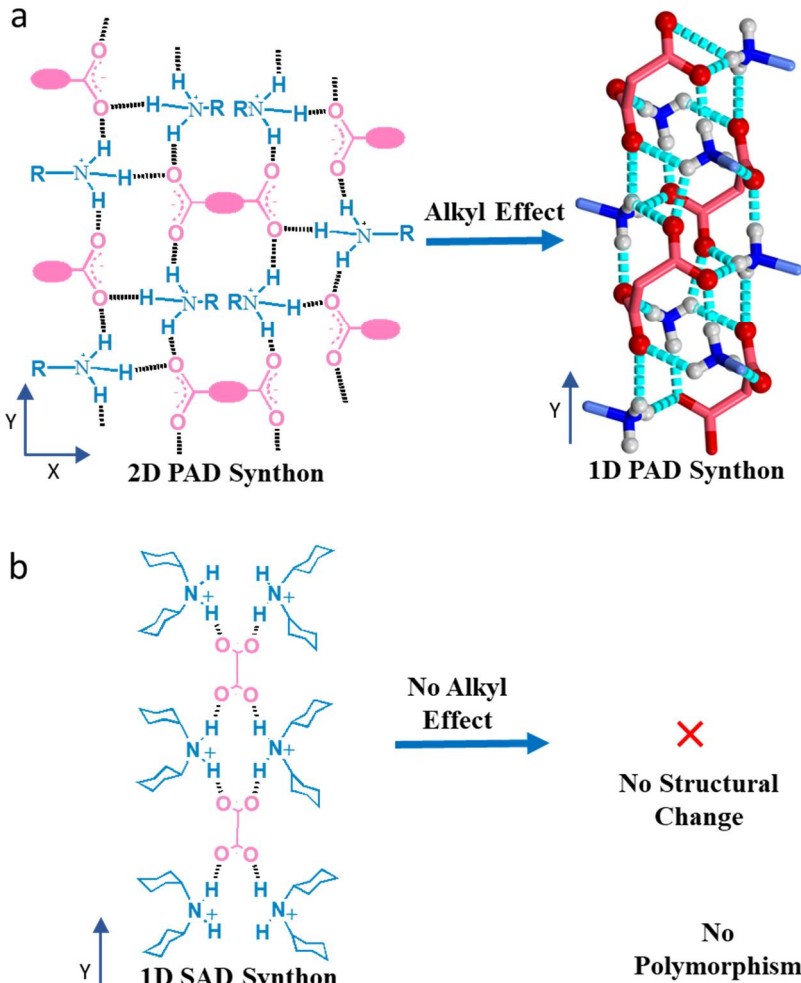

**Figure 1.** Alkyl effects on dicarboxylate salts. (**a**) Van der Walls's interaction at an increased alkyl chain in the amine part transforms the 2D Primary Ammonium Dicarboxylate (PAD) synthon to 1D synthon. The backbone of a dicarboxylic acid, marked by pink ovals, acts as a linker between the two parallel 1D hydrogen-bonded network (*Y* axis) to propagate an orthogonal growth towards the *X*-axis in building an overall 2D synthon. After increasing the carbon chain length at the primary amine part, the alkyl group (light blue) shows the lamellar effect with 360° orientation around the 1D synthon and bends the carboxylic acid backbone to generate the tubular structure. (**b**) No alkyl effect was found on the robust 1D Secondary Ammonium Dicarboxylate (SAD) synthon.

When in a charge transfer complex, one component does not melt up to 350 °C and undergoes crystalline phase changes that may impart heat-set gelation [23]. In a charge transfer (CT) complex derived from a pyridyl-ended NDI-based acceptor and pyrene-based donor, the acceptor does not melt at 350 °C and, basically, this component does not allow the CT complex to be dissolved at the hot stage in the solution and the gelling CT functionality [24] and the long-chain [25] induces gelation.

## 2. Designing Strategy of a Heat-Set Gelator by a Crystal Engineering Approach

When the gelator is heated in a large amount of solvent, it forms a highly viscous semi-solid material, termed heat-set gel or reverse thermal gel. With decreasing temperature, the system converts into a colloidal solution. A small amount of gelator can give

rise to a heat-set gel far above the boiling point of the gelling solvent [6]. The five combinatorial libraries of PAD salts were prepared using a crystal engineering approach to study their structural properties in displaying gelation abilities [6–10]. By investigating their temperature-dependent crystalline phase changes, it was noticed that the dicarboxylic acids with higher melting points produce the heat-set gel as PAD salts [6] where three principles can be derived. The dicarboxylic acids used (oxalic acid, maleic acid, tartaric acid, phthalic acid, and isophthalic acid) melt below 200 °C and dissolve in MeOH while preparing PAD salts, but they cannot form a heat-set gel even though the same series of amines (dodecyl amine to hexadecyl amine analogs) are used in preparation [8]. From the observation of PAD salts, the following hypotheses are made:

(a) The dicarboxylic acids with a high melting point (>350 °C) and poor solubility against organic solvent play important roles in designing heat-set gelators. The acids which are insoluble in solvents produce insoluble salts and can not even be dissolved at a high temperature. This feature helps form the Self Assembled Fibrillar Networks (SAFiN) in the solid state to immobilize the gelling solvents.

(b) The primary amines selected here in producing the gelators will not allow the gelators to melt or dissolve below the heat-set gelling temperature, ideally above 350 °C. In addition, the amines should form a weak van der Waals interaction which basically stabilizes the 1D HBN and 2D HBN in different temperature sets.

(c) The gelator structure should exhibit a temperature-dependent crystalline phase transition, where the higher temperature phase should promote the self-assembling of the gelator molecule, and the lower temperature phase should break it [6]. However, the lower-temperature crystalline phase can also form gels in different solvents.

Generally, amide, urea, azo, ferrocene, etc. backbones in dicarboxylic acid would introduce the ideal characteristics for deriving the heat-set gelling property (Figure 2). In another report, an amide moiety-driven dicarboxylic acid, after reacting with primary alkyl amine, also produced a series of PAD synthon-based heat-set gels [9]. The backbone responsible for gelation or any other functional groups present in the gelator molecules should not compete with carboxylate or ammonium functionalities to generate any competitive synthons [26]. When dicyclohexyl amine was reacted with 3,3′-ureabenzenedicarboxylic acid or 4,4′-ureabenzenedicarboxylic acid, the product never produced the SAD synthon. Therein, the urea functionality (-NH-CO-NH-) forms a stronger hydrogen bond than the conventional ammonium carboxylate bond. Any additional competitive hydrogen bonding sites at the gelators may appear desirable for supramolecular synthon in gel formation.

Two combinatorial libraries derived from ferrocene dicarboxylic acid and azobenzene-4,4′-dicarboxylic acids show the heat-set gelling property in a series of organic solvents (12, 13). The 5.0 wt % DMSO gel of hexadecyl ammonium azobenzene-4,4′-dicarboxylate (Figure 3a) exhibits clear two-phase transitions in the DSC study (the heating and cooling rate was 10 °C per minute (Figure 3b). The failure of Secondary Ammonium Dicarboxylate synthon in producing a heat-set gel must be addressed here to further illustrate the usefulness of the PAD synthon despite having azobenzene-4,4′-dicarboxylic acid or ferrocene dicarboxylic acid [13,14]. SAD synthons were produced by combining dicarboxylic acids with secondary amines, but none of them exhibited the heat-set gelling property [13,14]. The structural robustness never allowed the SAD synthon-based salts to form different crystalline states. Therein, the salts never changed their crystalline phases even by changing the temperature, preventing them from displaying heat-set gelling properties.

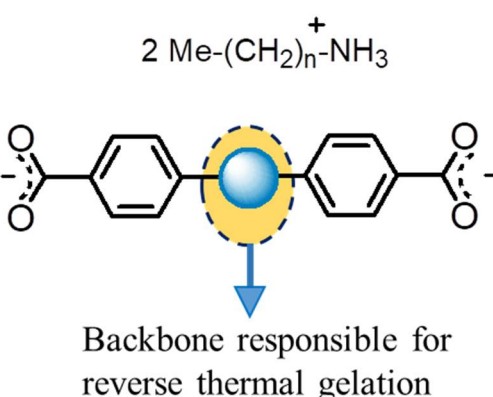

**Figure 2.** The schematic presentation of a dicarboxylic acid that can show heat-set gelling behavior. When a specific backbone such as azo, amide, urea, or bipyridine is used, the resultant dicarboxylic acid would be generally poorly soluble in methanol and does not exhibit the same melting point; they generally decomposes at around 350 °C.

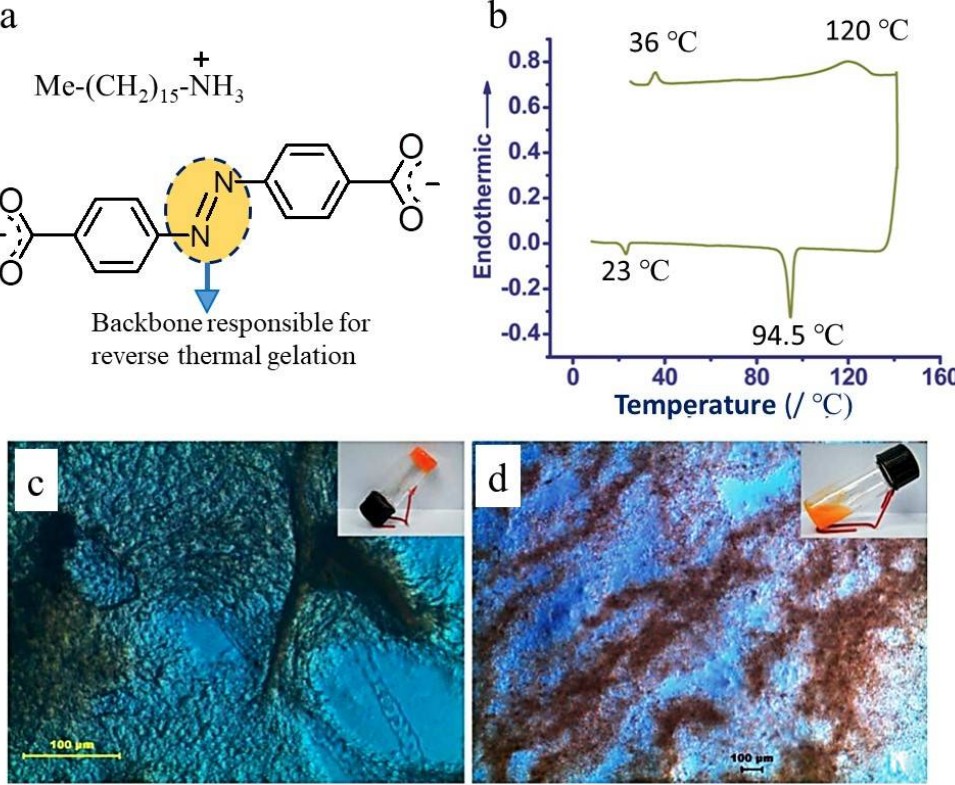

**Figure 3.** Reverse thermal gelling behavior of dicarboxylate salts. (**a**) Heat-set gel derived from hexadecylammonium azobenzene-4,4′-dicarboxylate. (**b**) DSC of 5.0 wt % DMSO gel (the heating and cooling scan was 10 °C min⁻¹). (**c**) HOPM of the semitransparent heat-set gel, made with chloro-benzene (bp 132 °C); the image was taken at 180 °C in an open atmosphere (Scale bar 100 μm). (**d**) Formation of colloid during a cooling cycle at room temperature (Scale bar 100 μm) ((**b–d**) adapted with permission from Ref. [6]. 2012, Royal Society of Chemistry).

When we use dicarboxylic acid, it does not melt on heating, rather, it charred around 350 °C, displayed very poor solubility in the heat-set gelling solvent, and would not melt/dissolve to form a solution (sol) at its salt states. The acid did not show the solid-liquid phase change at a higher temperature, rather it decomposed. This property is also reflected in the salt form, the salts produced from such dicarboxylic acids also do not dissolve in the heat-set gelling solvents. In the beginning, the mixture of a gelator

and its corresponding salt does not produce a clear solution (sol) on heating but heating the mixture for a longer time transforms the opaque colloid into a semitransparent gel. (Figure 3c). The cooling of such a heat-set gel will again produce colloid (Figure 3d). However, when a gelator melts at a higher temperature or is dissolved in the gelling solvent at its corresponding bp, it can never form the fiber necessary to trap the solvent in the formation of heat-set gels.

## 3. Heat-Storing Efficiency

Heat-set gels form in the heating cycle and become colloids while cooling. All the reversible heat-set gels are comprised of two distinct molecular packing patterns. At higher temperatures, they form a self-assembly, and at a lower temperature, they form a colloid. In some cases, the lower temperature crystalline phase may impart gelation in some different solvents. Primary ammonium azobenzene-4,4′-dicarboxylate salts exhibit gelation at DMF/DMSO but impart gelation at a higher temperature in aromatic solvents (Ph-Me, PhCl, xylene, etc.) [6]. The crystalline packing pattern of these PAD salts changes under colloid-gel phase transition. The heat-set metallogel sometimes shows the transition in coordination geometry of the metal ion ($T_d \leftrightarrow O_h$) moving from the sol to gel state while heating [15]. Gelation is a frustrated crystallization process. The gelator molecules are not exactly ordered as in a crystalline phase but are organized better to an amorphous form. The simulated pattern, with higher intensity produced from single-crystal X-ray diffraction data, often fits the powder X-ray diffraction pattern of gel/xerogel, explaining gelator molecules grouped similarly to the crystalline state. Thus, gel phases can also absorb thermal energy for randomly disorienting the self-assembled gelator molecules and exhibit an endothermic peak in a differential calorimetric study.

Due to the metastable crystalline orientation in the heat-set gel phase, molecules will not be reluctant to reorganize again in a different crystalline state or disorganize themselves into forming colloids. Thus, the heat-set gel will not release energy spontaneously during the cooling process. Several weak supramolecular bonds urge the metastable state to remain claim for a while and enhance the specific heat property of heat-set gel. The supramolecular packing and their metastable phase will make the heat-set gel reluctant to release heat and simultaneously enhance the specific heat capacity. The high-temperature gel phase (heat-set gel) and the colloid phase can be characterized by employing rheology. In rheology, the storage modulus (G′) and the loss modulus (G″) will be higher in comparison to the colloid state. This is a reverse observation of a normal thermo-reversible gel. In the food industry, Alaska Pollock (one kind of fish) is used in preparing surimi paste (East Asian food) [27]. Surimi paste also forms a heat-set gel in a water medium and shows phase transition under the DSC study. The presence of an endothermic peak around 65 °C ensures the phase transition from a heat-set gel to a colloidal state. After measuring the specific heat at different temperatures, it was observed that, after 65 °C, the specific heat falls sharply. The temperature-dependent rheology demonstrates that the visco-elastic nature falls steadily around 65 °C.

## 4. Introducing Nucleation Functionality in Gelator Molecule in Growing API Crystals

Heat-set gels can be exploited in crystallizing pharmaceutical compounds, as this medium can sustain heat at higher temperatures and the gel network creates a large surface area for nucleation. Crystallization temperature plays a crucial role in influencing nucleation processes which are driven by the supersaturation of the sample [28]. An increased temperature leads to a decreased supersaturation in most cases, increases solution conformational mobility, and makes a more dominant entropic contribution to the free energy of the nucleus. The nucleation of entropically stable, high-temperature phases is predicted to be favored by these circumstances; more traditional solid form screening tests may never find such forms. As a result, a new avenue for crystallizing a series of polymorphs will open up, potentially leading to the generation of a series of bio-active polymorphs. In addition to the gelling solvent and gel temperature, the gelator structure

also plays a crucial role in crystallizing the guest molecules (APIs) (Figure 4) [29–31]. The isolation of a single crystal or polycrystalline material will be facilitated by simply reducing the temperature, with the result of the dissipation of the gel, to give a colloidal solution. This high-temperature gel phase crystallization represents a completely novel paradigm in the crystallization of organic compounds such as pharmaceuticals and agrichemicals.

the·ammonium·group·and·the·drug·mimic.¶

**Figure 4.** Structures of active pharmaceutical ingredients (APIs). The aromatic rings and alkyl part of these APIs can promote template-induced nucleation at the aromatic part (in acid) or alkyl chain (in amine part), respectively.

The match between the parent and daughter phases does not have to be particularly close; only the 'template' part needs to match one face of the daughter crystal nucleus [28]. Crucially, therefore, there is every reason to expect that a gel fiber could nucleate a particular polymorph [30,31] of a pharmaceutical. If the API can bind to a SAFIN in such a way that the short-range periodicity information within a gel fibril can match up with, and hence be transferred to, a periodic nucleus of a particular pharmaceutical solid form, chemically tailoring will be introduced to promote a 'templated' nuclei formation from the SAFIN backbone to overcome the slow nucleation rate in the gel medium [3]. It was witnessed that the solvent generally controls the solute-binding functionality in gel fiver [32,33]. Generally, solvophobic effects dominate in polar media, particularly water, and hence, it is appropriate to tailor the hydrophobic regions of the drug within a hydrophobic long-chain alkyl residue. In response, this hydrophobic part will be used as a template for the nucleation of the drug molecules in a reverse thermal gel medium. Drug-like functionality based on representative drug substances such as ROY, linezolid, carbamazepine, isoniazid, ritonavir, and mesalamine can be introduced into the alkylammonium chain. In a typical case, the arylamine functionality in the multiple myeloma drug lenalidomide can be directly connected to the end of a ω-functionalized protected alkyl amine. Another option consists of its incorporation as the ammonium component of the PAD or giving a drug-moiety terminated ammonium component with an oligo ethylene spacer between the ammonium group and the drug mimic.

*Precaution of Crystallization in Heat-Set Gel*

Regarding the side reactions with gelators and solvothermal degradation of APIs, their thermal, physical, and chemical stability should be concerned during heat-set gel-

mediated crystallization. The side reaction can take place if the gelator and the guest molecule (targeted for crystallization) can form any competitive synthons [26]. As well as the additional competitive functional groups, organic salts similar to primary ammonium monocarboxylate (PAM) [34,35], secondary ammonium monocarboxylate (SAM) [36], or secondary ammonium dicarboxylate (SAD) [13,14] can undergo an exchange reaction and result in a decomposing heat-set gel.

In addition to polymorphic form(s), temperature can also have an effect on crystal habit and hence offers a means to modify unfavorable crystal morphology and understand the morphological outcomes. Thus, after forming crystals in a certain gel at a definite temperature, the polymorphism should also be checked. Supersaturation will be generated by thermally cycling to a high temperature to just dissolve all of the substrate material present and then cooled to a temperature within the gel range of the system. The concentration of APIs must be optimized at a given condition. The crystals made at a higher temperature would not be susceptible to ordinary thermal shock. However, their solubility in water is the main concern, as, after all, they are to be administered to the human body. Moreover, slow evaporation of the solvent at the high-temperature gel phase can also increase supersaturation to promote nucleation. Thus, the evaporation rate of the gelling solvent must be controlled, otherwise, the gel can be dried before APIs crystallize. This reverse thermal gel can be decomposed into a colloidal state by cooling the system slowly to recover the drug substance by filtration. This simple method of purification is a key aspect arising from the nature of heat-set gels. However, given the frequent occurrence of template-assisted crystal growth, the gel fiber should always be thoroughly removed from the API crystals.

## 5. Futuristic Use of Heat-Set Gel

A few years ago, a reverse thermal gel was employed for open fetal surgery on spina bifida defects [37]. However, as these types of engineered heat-set gels can sustain the gel state above the BP of the gelling solvent, the gels can be used for different purposes in developing futuristic chemical technology [38].

### 5.1. Crystallizing Bioactive APIs

A major application of heat-set gel would be as a novel crystallization medium. Growing pharmaceutical crystals in a gel medium are often advantageous as nucleation can be started on the gel fiber [1]. However, the heat-set gel can break the limit of crystallization temperature, which can be above 200 °C, even as the gels are stable above 200 °C. Thus, the heat-set gels can be used as a unique high-temperature medium for carrying out heat-set gel-mediated crystallization under different temperatures and gelling solvents in developing the polymorphs. Crystallization in a heat-set gel medium for targeted pharmaceutical compounds at a sustained high temperature is highly underexplored. Noticeably, crystallization temperature plays a crucial role in influencing the nucleation processes which are driven by the supersaturation of the sample. Therefore, an increased temperature leads to decreased supersaturation in most cases, increased solution conformational mobility, and a more dominant entropic contribution to the free energy of the nucleus. These factors are expected to promote the nucleation of entropically stabilized, high-temperature phases. Such forms may never be discovered by more conventional solid form screening experiments.

By introducing the APIs in a different crystalline phase, the bioactivity or water solubility can be enhanced, improving their effectiveness as medicine [39,40]. Gels are known to have been developed into pharmaceutical polymorphs in the last decades and the selective preparation of polymorphs is extremely important to ensure the purity of APIs [41,42]. Polymeric microgels with a tunable microstructure can be employed for developing selective polymorphs [43]. A bis(urea) gelator was designed to specifically mimic the chemical structure of the highly polymorphic drug ROY [44]. By chemical tailoring over the gelator backbone, the pharmaceutical compounds will be crystallized [45]. Polymorphism on organometallic drugs such as cisplatin was developed in a supramolecular gel medium [31].

Exploiting this concept, a large number of organic APIs (e.g., ritonavir, phenylbutazone, paracetamol, carbamazepine, cimetidine, ROY [44], and linezolid) can be added in excess to the gel medium in the low-temperature solution form and heated gently in order to dissolve the compound and induce gel formation. Some of these compounds are known to exist under varied polymorphic forms. In addition to polymorphic form(s), the temperature can also have an effect on the crystal habit and, hence, offers a means to modify unfavorable crystal morphology and understand the morphological outcome. Supersaturation will be generated by thermally cycling to high temperatures to dissolve all of the substrate material presents and then cooled to a temperature within the gel range of the system. This reverse thermal gel can be decomposed into a colloidal state by cooling the system slowly to recover the drug substance by filtration.

*5.2. Thermal Energy Storage System*

In addition to crystallization, the gel can harvest thermal energy; by physically squeezing the gel, it releases the heat within a few seconds to come to the natural boiling point of the gelling solvent. The heat-set gels are quite stable at more than 100 °C above the BP of gelling solvent. Just by squeezing the gel with a spatula, the gel releases the gelling solvent outside of its 3D network structure. Under an open atmosphere, the solvent releases the extra heat immediately and comes to the natural boiling point. Thus, within about a couple of seconds, it releases the additional heat into the atmosphere. This technique can also be used for sudden heat thrust to a certain system. If not disturbed, the gel can release heat slowly and the amount of the heat would be far higher than ordinary gels. A convenient specific heat capacity in the heat-set gel can be implemented in developing air conditioning and adsorption refrigeration technology. For such a purpose, the heat-set gels should possess a lower gelling temperature, the gel should be derived from water, methanol, or ethanol kinds of solvents, and, finally, the gel should absorb the unwanted gas molecules. Water seized silica gel has shown such properties, but the gel shows a convenient specific heat of 65 °C, which must be lowered [46].

**6. Conclusions**

Reports on heat-set gels are so far serendipitous and there is no design strategy for preparing a new heat-set gel with a certain purpose. As heat-set gels are highly unexplored for developing pharmaceutical polymorphs inside the gel medium, a novel designing strategy has become important for growing polymorphs. A series of heat-set gelators can be synthesized by exploiting Primary Ammonium Dicarboxylate synthon, where the dicarboxylic acids are insoluble in the gelling solvents and cannot be melted below 350 °C. If the dicarboxylic acids become soluble and can be melted below 350 °C, despite showing the heat-set gelling property, the resultant salts will exhibit a thermo-reversible gelling property.

The heat-set gel medium can maintain a certain elevated temperature for a longer time period under an open atmosphere and can be used for crystallizing the high-temperature conformation of drug molecules to enhance bioactivity. As molecular conformation changes with temperature, the heat-set gel can seize the conformations of API at a particular temperature through crystallization to enhance bioactivity. For such an objective, a heat-set gel designing strategy is hypothesized here for the first time which can also be extended in developing different heat-storing technologies or preparing high-temperature mediums for organic reactions. The increased specific heat property in the heat-set gels results from the crystalline metastable molecular arrangement at the higher temperature range, making the heat-set gel state reluctant to release heat when cooling in comparison to the regular supramolecular gels. Such higher specific heat characteristics in heat-set gels can even stabilize the gelling solvents beyond their corresponding boiling point under an open atmosphere and can release the thermal energy within a few seconds just by squeezing the gel to produce a heat thrust.

**Funding:** This research received no external funding.



**Data Availability Statement:** Not applicable.

**Conflicts of Interest:** The authors declare no conflict of interest.

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
