# Peer review of "Designing Heat-Set Gels for Crystallizing APIs at Different Temperatures: A Crystal Engineering Approach"

_2305-7084, doi:10.3390/chemengineering6050065_

Round 1

Reviewer 1 Report

This Perspective describes designing strategy of heat-set gelator based on APIs. Since the approach described in this manuscript is useful for studies on self-assembled supramolecular systems, I consider the paper acceptable for publication at ChemEngineering after addressing the comments and thorough revision of the manuscript.

Comments

1) General gelators melt when heated and form gels when cooled. I think the introduction should properly explain and emphasize the differences (behavior and characteristics) between heat-set gels and general gels.

2) Figure 1a illustrates the effect of alkyl on the supramolecular structure of dicarboxylates, but seems ambiguous and confusing. In other words, the mode of hydrogen-bond network formation seems to be emphasized, making it difficult to understand the effect of alkyl on molecular packing. Is it possible to illustrate the influence of alkyl a little more clearly?

3) The figure number on line 214 should be Fig. 4, not Fig. 3. In addition, please check and correct the numbering of the figures in the text. For example, Fig. 2a on line 128 should be Fig. 3a.

Author Response

Comments and Suggestions for Authors

This Perspective describes designing strategy of heat-set gelator based on APIs. Since the approach described in this manuscript is useful for studies on self-assembled supramolecular systems, I consider the paper acceptable for publication at ChemEngineering after addressing the comments and thorough revision of the manuscript.

Response. I sincerely thank the reviewer for his kind comments and constructive suggestions. I have modified the manuscript accordingly.   

Comments.

1) General gelators melt when heated and form gels when cooled. I think the introduction should properly explain and emphasize the differences (behavior and characteristics) between heat-set gels and general gels.

Response: I sincerely acknowledge the reviewer for this important issue to bring to my notice. I have carefully modified the introduction part and I think, now it becomes clear.

2) Figure 1a illustrates the effect of alkyl on the supramolecular structure of dicarboxylates, but seems ambiguous and confusing. In other words, the mode of hydrogen-bond network formation seems to be emphasized, making it difficult to understand the effect of alkyl on molecular packing. Is it possible to illustrate the influence of alkyl a little more clearly?

Response: This is a very important figure and it was not up to the publication standard. I have modified it according to the suggestion. I thank the reviewer for his valuable advice.

3) The figure number on line 214 should be Fig. 4, not Fig. 3. In addition, please check and correct the numbering of the figures in the text. For example, Fig. 2a on line 128 should be Fig. 3a.

Response: I thank the reviewer for this suggestion. All the figure numbers are now mentioned in properly. I  am extremely thankful to the reviewer for providing me his valuable suggestions and an additional opportunity to modify the manuscript for publication. 

Reviewer 2 Report

The current paper studies heat-set gel designing strategy. The study sounds interesting, however; the paper structure is hard to follow. And the application of such material has not been well-specified and studied. References are quite old as well. The author is encouraged to cite more recent studies in the same topic. Below are also some detailed comments:

1-      The Introduction lacks enough support for the current study. It should be improved!

2-      The paper has no experimental section!

2-      Figure 1 is vague. The author should show the mechanism of interactions. What do the round pink balls mean in Figure 1a?

3-      Figure 1 should be referred in the main text. Unfortunately, the author forgot to refer to Figure 1 in the main text.

4-      Figure 2 is not adding any value to the discussion. The author should revise it in a way that it complements the discussion.

5-      The author mentioned Fig 2a and Fig 2b in the main text. However, Figure 2 is just a single structure. The author must check the figure numbers.

6-      Figure 3c and 3d have not been referred in the main text. The author must make it clear that what the point of each figure is in the paper!

7-      There are two Figure 3! It is not clear what the author is trying to discuss!

In conclusion, it looks like that the paper needs several back-and-forth to be get ready for publication consideration. At this point, I recommend returning the paper to the author for major restructuring and rewriting!

Author Response

Comments and Suggestions for Authors

The current paper studies heat-set gel designing strategy. The study sounds interesting, however; the paper structure is hard to follow. And the application of such material has not been well-specified and studied. References are quite old as well. The author is encouraged to cite more recent studies in the same topic. Below are also some detailed comments:

Response: I thank the reviewer for his kind suggestions and for providing me the opportunity to modify the manuscript. I have modified the manuscript according to his very crucial suggestions and comments. Regarding the reference part, I have cited almost all important heat-set gel papers to date. Reports on serendipitous findings are still not becoming routine work, and a good research pulse has not yet been established, because there is no available heat-set gel design concept. I think this modified version will be able to address his all concerns.

1-      The Introduction lacks enough support for the current study. It should be improved!

Response: The introduction part is now modified with the utmost care. Based on all available reports on supramolecular heat-set gels and related supramolecular synthon, I have modified the text. I thank sincerely the reviewer for his kind note.

2-      The paper has no experimental section!

Response: As this is a perspective paper based, here a new concept is built based on some previous reports by various groups, it does not contain the experimental section. I selected the ‘manuscript type’ as a ‘perspective’ and now I changed the type at the manuscript from ‘article’ to perspective’. I am very much thankful to the reviewer for his kind remark.

2-      Figure 1 is vague. The author should show the mechanism of interactions. What do the round pink balls mean in Figure 1a?

Response: Figure 1 and its caption, both are now modified properly. The pink balls represent the structural part, that imparts insolubility in the gelling solvent. I have mentioned it also in the figure caption. I sincerely thank the reviewer for his nice suggestion.

3-      Figure 1 should be referred in the main text. Unfortunately, the author forgot to refer to Figure 1 in the main text.

Response: All the figures were not referred at the main text. I have modified them. I thank sincerely the reviewer for bringing this important issue to me.

4-      Figure 2 is not adding any value to the discussion. The author should revise it in a way that it complements the discussion.

Response. The text was not properly written and thus the figure became irrelevant. Now I have modified the text and I hope, this time the reviewer will be satisfied. I thank the reviewer sincerely for his minute observation and valuable suggestions.

5-      The author mentioned Fig 2a and Fig 2b in the main text. However, Figure 2 is just a single structure. The author must check the figure numbers.

Response. These would be figure 3a and 3b. I have modified them accordingly. I am extremely thankful to the reviewer for his nice suggestions.

6-      Figure 3c and 3d have not been referred in the main text. The author must make it clear that what the point of each figure is in the paper!

Response. I have modified the text accordingly. I sincerely thank the reviewer for bringing the issue to me.

7-      There are two Figure 3! It is not clear what the author is trying to discuss!

Response: I have carefully modified all the problems. I hope this present format of the manuscript will the satisfactory to the reviewer. I am very much thankful to the reviewer for his great suggestions and precious comments.

In conclusion, it looks like that the paper needs several back-and-forth to be get ready for publication consideration. At this point, I recommend returning the paper to the author for major restructuring and rewriting!

Response: I am extremely thankful to the reviewer for his remarks and suggestions. I hope this revised version will cover his concerns. In case the manuscript still needs to revise, I shall be very happy to address his concerns again.

Round 2

Reviewer 1 Report

The manuscript has been properly revised. The revised manuscript will be accepted for publication without alterations.

Reviewer 2 Report

The author addressed all the comments.